# Trypanosomatid research in Brazil: a systematic analysis of regional and temporal trends

**Bárbara Marinho, Izabela Mamede, Júlia Raspante Martins, André Rodrigues, Ana Gabrielle Batista de Melo, Adalberto Sales Miranda-Junior, Alice Rios Neto, Amanda Carolina da Silva Nunes, Bruno Carvalho Resende, Dáfne Oliveira, Darlan Oliveira da Silva, Frederico Gabriel de Carvalho Oliveira, Jéssica Duarte, Lorrane Diniz de Carvalho Silva, Wesley Roger Rodrigues Ferreira, Daniela De Laet-Souza, Andrea Mara Macedo, Glória Regina Franco/+, Carlos Renato Machado/+**

Universidade Federal de Minas Gerais, Instituto de Ciências Biológicas, Departamento de Bioquímica e Imunologia, Laboratório de Genética Bioquímica, Belo Horizonte, MG, Brasil

**BACKGROUND** Trypanosomatid infections such as Chagas disease (CD) and leishmaniasis remain major public-health concerns. Brazil has a long tradition in this field, yet a consolidated, country-level view of outputs, impact and collaboration patterns is useful to guide scientific policy.

**OBJECTIVES** To characterise Brazilian scientific production on *Trypanosoma cruzi*, *Leishmania* and *Trypanosoma brucei* (2010-2021), describing temporal trends, regional contributions, collaboration networks and journal impact.

**METHODS** We performed a bibliometric analysis of PubMed records retrieved with Medical Subject Headings (MeSH) for each pathogen/disease pair, covering publications from 1 January 2010 to 31 December 2021 (search date: 21 July 2022). Data items included article type, year, journal, author affiliations (countries/institutions) and, for Brazil, the geographical region of the corresponding author. Descriptive statistics and visualisations were generated in R.

**FINDINGS** From 21,713 records, 6,478 were affiliated to Brazil. Brazil contributed a substantial share of the global literature, particularly for *T. cruzi* (≈40%) and *Leishmania* (≈30%). Within Brazil, output increased over time with growing participation from the north and northeast, alongside expanding inter-institutional and international collaborations. Most publications appeared in higher-impact journals (Q1/Q2), with recent gains in Q1 outputs in historically under-represented regions. Original research predominated over reviews across the period.

**MAIN CONCLUSIONS** Brazilian trypanosomatid research shows sustained growth, increasing regional dispersion and rising international engagement, with a strong presence in high-impact journals. Continued support for collaborative networks and equitable funding across regions could further enhance national and global impact.

Key words: *Trypanosoma cruzi* - *Leishmania* - Brazil - tripanosomatids research - bibliometrics

Research on neglected tropical diseases (NTDs) in Brazil, particularly Chagas disease (CD) and leishmaniasis, has historically received significant national attention due to the high burden of these diseases within the country. Leishmaniasis is endemic in several Brazilian regions, with over 27,000 cases of visceral leishmaniasis (VL) reported annually, accounting for approximately 90% of cases in the Americas.[1,2] Meanwhile, CD affects around 7 million people worldwide, with a significant prevalence in Latin America, including Brazil, where transmission remains a relevant public health issue.[3,4]

Brazil has gained global recognition for its scientific production related to NTDs, partly due to the legacy of pioneers like Carlos Chagas, who thoroughly identified the disease that received his name in 1909. The impor-tance of this research is underscored by the ongoing need to improve control and treatment strategies, given their epidemiological complexity and the existing gaps in therapeutic approaches.[5] Additionally, leishmaniasis, both cutaneous and visceral forms, continues to pose an epidemiological challenge in Brazil, with a wide geographical distribution and a lethality rate of around 8% for untreated visceral cases.[1,6]

Other trypanosomatids, such as *Trypanosoma brucei*, have been studied more extensively on an international scale, particularly in Europe, due to significant epidemics of sleeping sickness across equatorial Africa in the early 20th century. These outbreaks prompted European colonial administrations to prioritise research and control measures against the disease.[7]

Financial support: FAPEMIG, CNPq (by fellowships to BM, IM, AGBM, ASMJ and DDLS), CAPES (by fellowships to JRM, ACSN, BRS, FGCO, WRRF).
+ Corresponding authors: crmachad1967@gmail.com | ⓘ https://orcid.org/0000-0002-8724-3165 / gfrancoufmg@gmail.com | ⓘ https://orcid.org/0000-0001-5245-2365

**Handling editor:** Adeilton Alves Brandão | ⓘ https://orcid.org/0000-0001-5877-607X

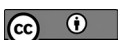

The progress of NTD research in Brazil has been significantly influenced by public policies and funding initiatives. The inclusion of NTDs in the National Agenda of Health Research Priorities (ANPPS) in 2006, and its update in 2010, was a significant milestones which prioritised diseases like leishmaniasis and Chagas, driving investment in research in these areas.[8] Examples of this include increased funding from agencies such as National Council for Scientific and Technological Development (CNPq) and Coordination for the Improvement of Higher Education Personnel (CAPES), which directed specific resources to NTD studies, as well as the strengthening of collaborative research networks among national and international institutions.[9] This support has led to a growth in scientific productivity, with a direct impact on public health policies and the training of specialised researchers.

The goal of this study is to analyse trends in Brazilian scientific production on *Trypanosoma cruzi* and *Leishmania*, identifying factors influencing this output, such as funding policies and regionalisation of research. By understanding these dynamics, this work aims to contribute to the formulation of strategies that strengthen NTD research in Brazil, promoting a more equitable and effective distribution of scientific resources which is crucial for addressing the challenges of NTDs.

### MATERIALS AND METHODS

This study was conducted to evaluate the trends in the publication of scientific articles related to the three pathogenic trypanosomatids — *T. cruzi*, *Leishmania*, and *T. brucei* — over the last 12 years (2010-2021).

*Inclusion criteria* - Included in this review were studies focused on scientific articles published related to the three pathogenic trypanosomatids: *T. cruzi*, *Leishmania*, and *T. brucei*, within the period from 2010 to 2021 [Supplementary data (Fig. 1)]. Were included: original articles, reviews, clinical trials, case reports, observational studies, randomised controlled trials, multicentre studies, and meta-analyses. No language restriction was applied. Comments, letters to the editor, editorials, and errata were excluded due to their nature as opinion pieces. Articles with incomplete author information or without a digital object identifier (DOI) were also excluded.

*Search strategy* - Searches were conducted in the PubMed database using specific Medical Subject Headings (MeSH) terms for each pathogen and their associated diseases: *T. cruzi* and CD, *Leishmania* and leishmaniasis, and *T. brucei* and sleeping sickness. The search covered publications from January 1, 2010, to December 31, 2021. The search strategies were formulated to capture all relevant publications within the specified period. The search was performed on July 21, 2022.

*Study selection* - All retrieved records were independently evaluated by reviewers involved in data collection. Initially, titles and abstracts were screened for eligibility. Subsequently, the full texts of relevant articles were reviewed. In case of disagreement among reviewers, discussions were conducted to reach a consensus. Duplicate articles were removed based on DOI.

*Data collection* - Data was extracted from each study using a standardised format that included the following information: year of publication, type of article, country of the first author, country of the corresponding author, name of the journal, number of countries involved among the authors, number of different institutions in the article, and the Brazilian region of the corresponding author. Extraction was performed by 12 independent reviewers, and the collected information was reviewed by an independent author.

*Data items* - The collected data items included:

• *Study details*: reviewer identification, study ID, extraction date, study title, author name, year of publication, and journal of publication.
• *Study method*: objectives, study design (cross-sectional, cohort, or clinical trial) or type of scientific output (original articles and reviews).
• *Primary and secondary outcomes*: proportion of publications related to each pathogen, prevalence of international cooperation, and geographical distribution of publications.

*Data analysis* - After data collection and extraction, all information was imported into R software (version 4.3.3), where all statistical analyses and visualisations were conducted. The R packages used for data manipulation and analysis were *dplyr* (version 1.1.2) and tidyverse (version 1.3.0).[10,11] Data visualisation was performed using the ggplot2 package (version 3.4.2).[12]

Extracted data was initially cleaned to remove any inconsistencies, such as undetected duplicates, incomplete entries, or entry errors. The *dplyr* package was used for filtering, grouping, summarising, and arranging data operations. Additionally, data consistency regarding publication years, types of articles, and author information was verified. An initial descriptive analysis was performed to characterise the volume of publications by pathogen (*T. cruzi*, *Leishmania*, and *T. brucei*) over the period from 2010 to 2021. Absolute and relative frequencies were calculated for each data category, including the type of article (original, review, clinical trials, etc.), country of the first author, and country of the corresponding author. Descriptive statistics were also calculated for numerical variables, such as the number of collaborating countries in each study and the number of institutions involved.

To examine trends over time, histograms were generated using *ggplot2*, showing the annual number of publications for each pathogen. These graphs were used to identify peaks and declines in scientific production, allowing a visual analysis of research trends for each organism. International cooperation was assessed by the number of countries and institutions involved in each publication. Stacked bar charts were used to visualise these collaborations, revealing patterns of collaboration in research on trypanosomatids. The geographical distribution of publications was analysed using thematic maps, highlighting regions with the highest scientific output. This analysis was conducted both in a global context and specifically for the regions of Brazil, aiming

to understand regional disparities in contribution to the scientific literature on the three pathogens. All figures and graphs were generated using *ggplot2* and were included to facilitate understanding of the analyses conducted. These graphs were exported in high resolution for use in the systematic review results. Detailed reports containing summarised tables and graphs were generated for each section of the analysis.

### RESULTS

*Systematic review results* - Search for MeSH terms of *T. cruzi*, *T. brucei*, and *Leishmania* from 2010 to 2021 resulted in a total of 21,713 articles, being 6,477 *T. cruzi*, 3,344 *T. brucei* and 11,892 *Leishmania*. 19,529 of the total were original articles, while 2,478 were reviews and 1,600 were others, like case report, clinical trial and Radonmised Controlled Trial. Of these articles, 6,478 were from Brazil and were further categorised by geographical regions and the specific Tritryps organisms studied.

*Brazil's publication ratio compared to the world and within its geographical regions* - Data analysis of publication ratio in *T. cruzi*, *T. brucei*, and *Leishmania* of Brazil compared to the rest of the world showed a stable increase with time (Fig. 1A). Over the past 11 years, although we have not observed major fluctuations in the distribution percentage, Brazilian research has shown little variation. Notably, during this period, Brazil was responsible for a major portion of global research related to *T. cruzi*, representing around 40% of all publications. Regarding *Leishmania*, Brazilian publications averaged approximately 30% over the last 11 years. However, concerning *T. brucei*, Brazil contributed a relatively smaller number compared to the global total. From this point onward, the analyses will focus on *T. cruzi* and *Leishmania* to show their publication trends inside Brazil.

Brazilian research publications on *T. cruzi* (Fig. 1B) reveal a trend of growth over time. However, it is evident that despite the significant increase from 2010 to 2021, there was a period of relative stability in 2012-2013. This was followed by a decline in 2015, which was reversed with an increase in the following years.

When analysing the geographical distribution of publications in Brazil (Fig. 1C), significant differences between the country's five regions stand out. The Southeast region leads in terms of average publications, while the Midwest region had the lowest average over the last 11 years, and the South region was the only that recorded a decrease. On the other hand, both

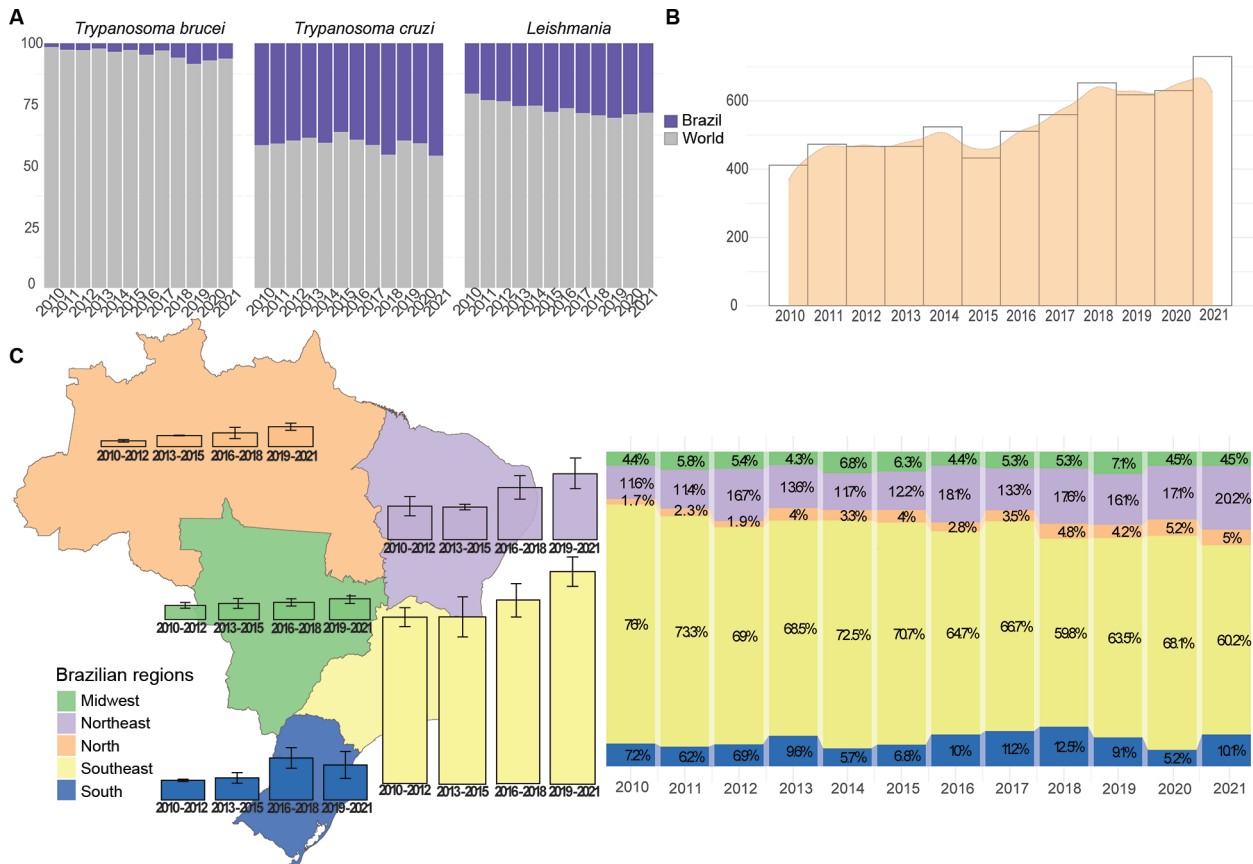

Fig. 1: (A) Percentage of publications in *Trypanosoma cruzi*, *Trypanosoma* and *Leishmania* per year in Brazil when compared to the rest of the world. On the x axis the years and on the y axis the percentage of replicates. In purple publications in Brazil and in grey, the rest of the world. (B) Number of publications in *T. cruzi* per year in Brazil using a barplot and a tendency curve in orange. (C) Dynamite plot using the mean of *T. cruzi* publications in three-year intervals in each of the five geographical regions of Brazil. Standard deviation among each three years is presented as a black line on top of the bars. On the side, stacked barplot with percent publication type per year. In orange, North; in purple Northeast; in yellow Southeast; in blue, South and in green Midwest.

the Northeast and the North regions had an increase in publication number. In the Northeast, the number rose from 43 in 2010 to 147 in 2021, while in the North, it increased from 17 in 2010 to 36 in 2021.

Furthermore, when considering each region's contribution to the total number of publications per year, we observed an increase in the proportion of publications from the north and northeast relative to the total, while the absolute increase in the number of publications in recent years, the Southeast region showed a percentage decrease. This geographic dispersion of research groups across Brazil aligns with trends in collaborative networks. From 2010 to 2021, there was an increase in the average number of affiliations per article nationally [Supplementary data (Fig. 2A)], reflecting more dispersed collaboration. Regionally [Supplementary data (Fig. 2B)], this increase occurs in the Northeast region and, mainly, in the north. These results suggest that the geographic dispersion observed in publications is accompanied by institutional diversification, consistent with the expansion and diffusion of scientific groups involved in trypanosomatid research in Brazil.

*Publication type and organism distribution within Brazil* - The production of publications related to *T. cruzi* and *Leishmania* in different regions of Brazil over time (Fig. 2A) exhibits a similar variation to the general trend of the *T. cruzi* total publications per region. In all Brazilian regions, the highest publication number is of *Leishmania*. However, this pattern is only present from 2012 onwards. Before this period, in all regions, except the Northeast, publications in *T. cruzi* were greater. Among all regions of Brazil, the Northeast showed the greatest triennial variation in *Leishmania* publications, ranging from 178 in the period from 2010-2012 to 350 in the period from 2019-2020, an increase of almost 100%.

Regarding the percentage of publication types produced in Brazil on *Leishmania* and *T. cruzi* (Fig. 2B), a predominance of original articles was observed over the years, followed by review articles, which showed an increase from 9% to 11% of the total publication per year. The remaining publications like case report, clinical trial and Randomised Controlled Trial, were categorised as other article types. This distribution reflects original research as the main output article type of the Brazilian community, which is responsible for over 30% of all worldwide articles on *T. cruzi* and *Leishmania*.

It is noteworthy that the North, Northeast, and Midwest regions have a higher proportion of articles published with a thematic focus on *Leishmania*, primarily original articles, while the Southeast and South regions have a similar proportion of original articles on *T. cruzi* and *Leishmania* (Fig. 2C). The proportion of review and other article types

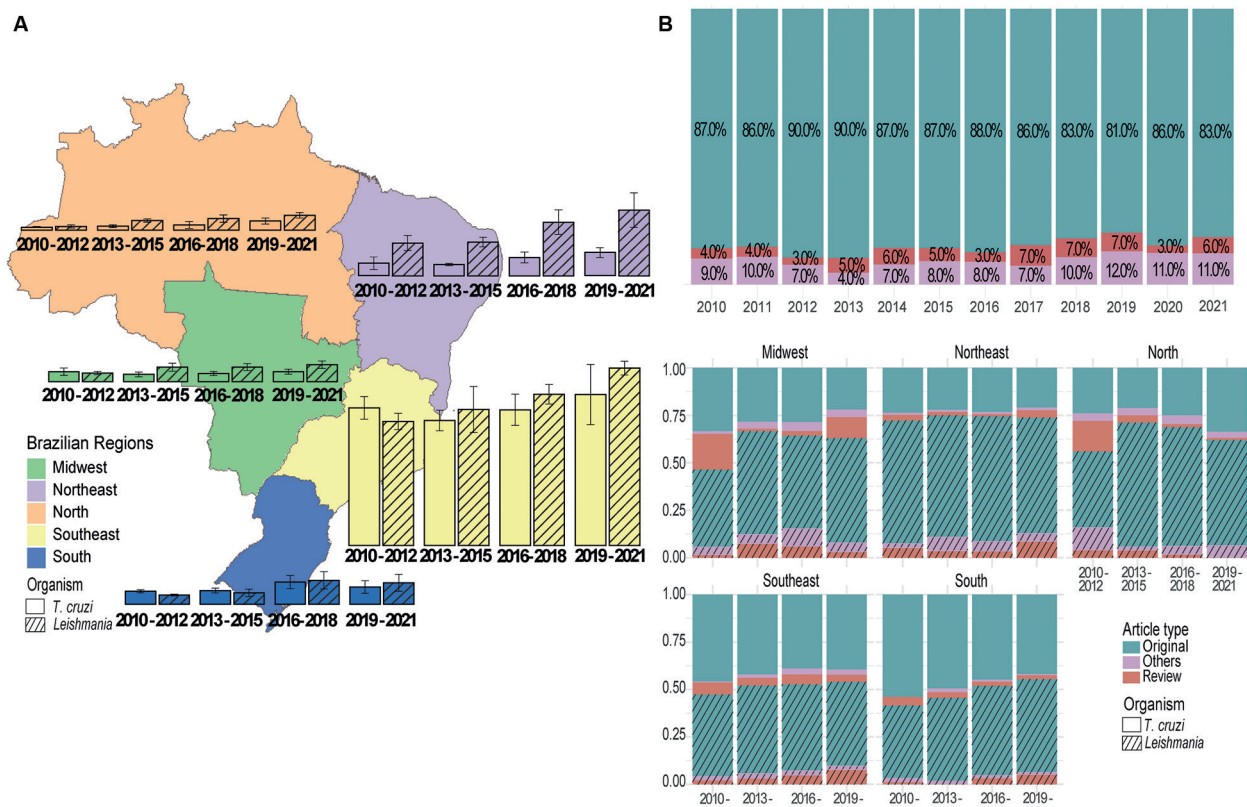

Fig. 2: (A) Mean of *Trypanosoma cruzi* publications when compared to *Leishmania* per year per region of Brazil. Dashed colours are *Leishmania* and solid colours are *T. cruzi*. Standard deviation is represented as a black line on top of the bars. (B) Barplot of the percentage per year of article types published in *T. cruzi* and *Leishmania* in Brazil. In blue, original articles, in green, reviews and in red other types of articles. (C) Barplot of percentage of article types published in *T. cruzi* and *Leishmania* per three-year intervals per geographic region in Brazil. Hashed colours are *Leishmania* and solid colours are *T. cruzi*. Colour scheme is the same as the letter A.

do not show a clear trend, but from the period between 2010-2012 to 2019-2021, there was an increase in original articles, with a significant decrease in reviews published on *T. cruzi* and *Leishmania*. However, for the Southeast and Northeast regions, despite the main article type being original, in recent years there is an increase in the number of published *Leishmania* review studies.

*Ratio of international collaborations among Brazilian regions* - A general growth in the number of collaborations is observed for both organisms and in all regions of Brazil. The Southeast region stands out as the region with the highest number of publications, increasing from an average of 1,181 articles between 2010 and 2012 to 1,707 between 2019 and 2021. However, this growth is not uniform across all regions. For instance, the Northeast region exhibits a progressive increase in collaborative publications over the years, particularly for *Leishmania*, which showed a more than two-fold increase in the number of publications from 2010-2012 to 2019-2021.

While for *T. cruzi*, there was a noticeable decrease in publications without collaboration in the Southeast region, indicating a shift towards more collaborative efforts over time. Meanwhile, no clear pattern of increase or decrease in non-collaborative publications was observed in other regions.

The increase in regional collaborations reveals notable disparities. While the Southeast maintains its position as the most collaborative region, the Northeast demonstrates significant growth in international partnerships, particularly in *Leishmania* research, highlighting its expanding role in global efforts to study the disease.

These results underscore the rising internationalisation of research on *T. cruzi* and *Leishmania* in Brazil. The Southeast continues to be the region with the most international collaborations. But the increase in international research on *Leishmania* in the Northeast reflects a growing network of collaborations and highlights the role of these partnerships in advancing scientific knowledge of these diseases.

*Impact of articles published by organisms and regions of Brazil* - Observing the percentage of publications among the Brazilian regions by quartiles (Figs 3-4), we observe that all regions of Brazil have more than 50% of all publications in Q1 (percentile scopus: 100-76) and Q2 (percentile scopus: 75-51) impact journals, for *T. cruzi* and *Leishmania*. This suggests a trend of high quality scientific publications on these parasitic organisms across all regions.

Throughout the period (2010 to 2021), there was a significant increase in publications in Q1 for *T. cruzi* and *Leishmania* in the North and Midwest regions, and exclusively for *Leishmania* in the Northeast region. On the other hand, the Southeast and South regions had a non-significant change in the proportion of publications in Q1 journals for both *T. cruzi* and *Leishmania* over the years.

Regarding publications in Q3 (percentile scopus: 50-26) journals, there was a tendency of decrease in *T. cruzi* publications, especially in the Midwest. As for the Q4 (percentile scopus: 50-26), there was an increase in publications of *T. cruzi* and *Leishmania* in the Midwest and South regions, as well as a decrease in the North and Northeast regions in the last three and six years analysed.

These results point to an increase also in the number of high impact publications, more notable in the North and Northeast regions, from 2016 to 2021. The Southeast region still remains as the most productive (Figs 2, 4) and producing the highest impact publications in trypanosomatids in Brazil. However, in recent years there was a significant increase in impactful research across various regions, reflecting a broader distribution of scientific contributions within Brazil.

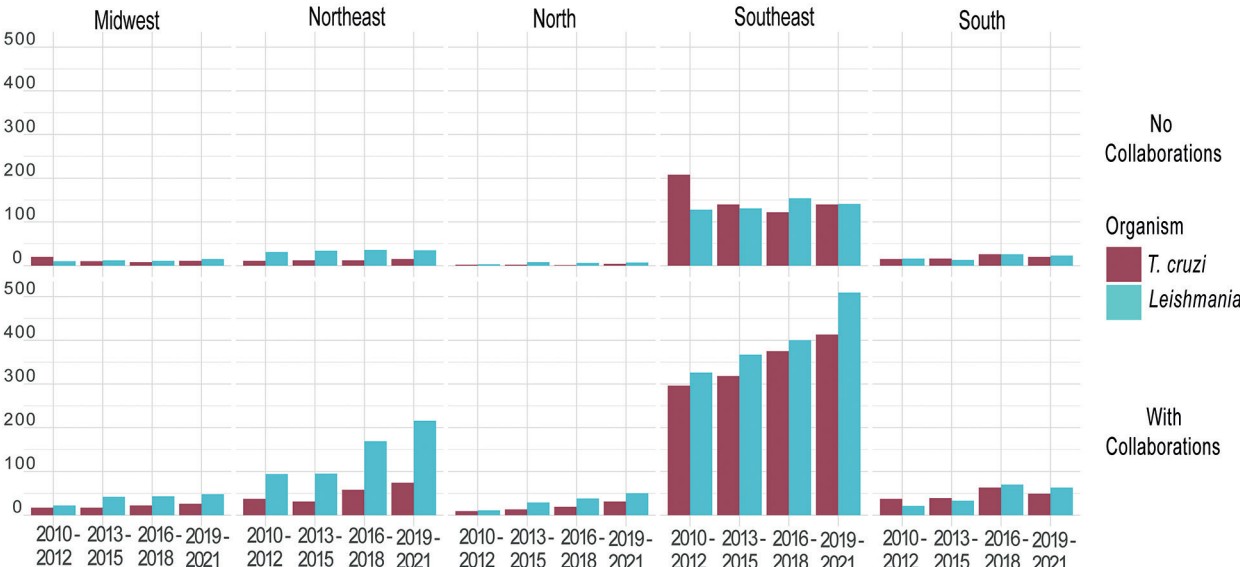

Fig. 3: bar plots displaying the total number of publications on *Trypanosoma cruzi* and *Leishmania* within three-year intervals across different regions of Brazil, indicating whether the publications involved collaborations with other research groups. In red information about *T. cruzi* and in blue, *Leishmania*.

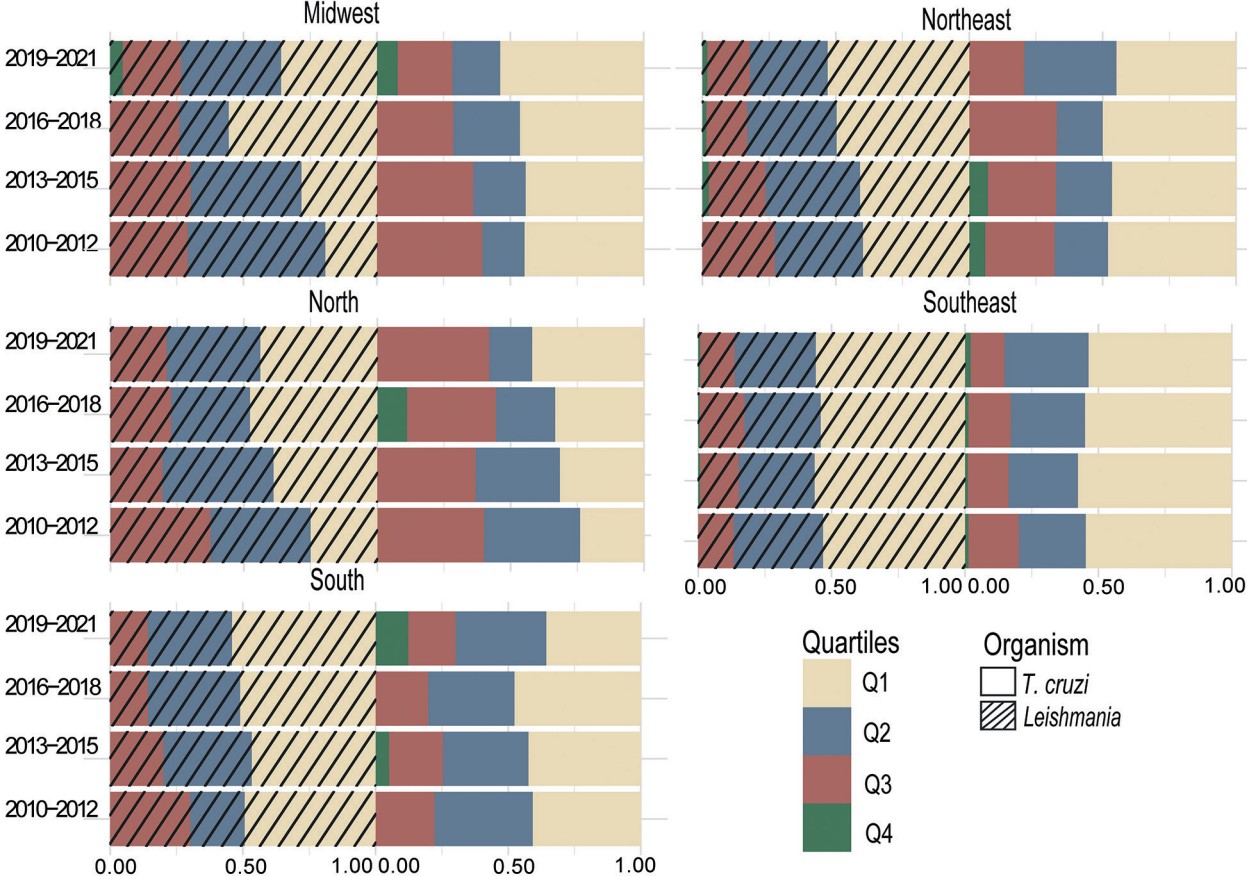

Fig. 4: stacked barplots of publication percentage of *Trypanosoma cruzi* and *Leishmania* per Brazilian geographical regions and separated by quartile of the journals. Hashed bars represent *Leishmania* and normal bars represent *T. cruzi*. In red, papers published in Q1 journals, in purple, papers published in Q2 journals, in teal, papers published in Q3 journals and in yellow, papers published in Q4 journals.

## DISCUSSION

Our results show a steady increase in scientific output on *T. cruzi* and *Leishmania* over the past 11 years, with Brazil making a significant contribution to global research, especially in *T. cruzi*, where it accounts for nearly 40% of global publications. However, there were periods of stability and decline, such as stability in 2012-2013 and a decline in 2015, followed by a subsequent recovery (Fig. 1A-B). Significant cuts in federal research funding between 2014 and 2022, which reduced available resources by approximately 60%, impacted the continuity and development of research projects, resulting in fluctuations in scientific output.[13,14] The decentralisation of scientific production in Brazil, with an increase in publications in the North and Northeast regions, contrasted with the traditional dominance of the Southeast region, which saw a decrease in its relative contribution. Regional incentive policies and greater inter-institutional collaboration facilitated this diversification.[14,15]

National policies have resulted in the spread of technical institutes and universities to other regions of Brazil. The Support Programme for Restructuring and Expanding Federal Universities (REUNI), established in 2007, is one of the initiatives within the Education Development Plan (PDE).[16] This programme resulted in the establishment of new federal university campuses and federal institutes of education, science, and technology in the North and Northeast region between 2002 and 2010.[17] REUNI contributed to the increase in the number of publications of north and northeast between 2010 and 2021 (Fig. 2A), as much of the scientific output in Brazil is driven by universities and graduate students. The implementation of scholarships in the northeast by CNPq was also significant. A census shows that, from 2002 to 2010, investment in scholarships and funding in the Northeast more than tripled (Source: CNPq - Census 2002-2010). However, investments have not been maintained in a consistent growth, and in 2002, the investment was 70% when compared to 55% in 2010.[18] The projects implemented by CAPES, CNPq, and REUNI, although not completely eliminating regional disparities in scientific production in Brazil, were crucial for the increase in original article publications (Fig. 2C).

Additionally, the support from funding state agencies was crucial during periods of federal cuts, allowing the maintenance of important projects.[11] The Southeast region of Brazil historically possesses the highest research throughput in the country. Moreover, the South and Southeast regions have the highest number of Graduate Education Programmes (PPGs).[19] These are a direct result not only of the colonisation and industry presence

during Brazilian monarchy and early republic but also has been maintained due to regional investment.[9] The new Growth Acceleration Programme (PAC), launched in 2023, promises significant investments in scientific research through 2026, offering hope for recovery and expansion of scientific infrastructure.[14] PAC will hopefully produce a broader geographical dispersion of research groups in Brazil, with states outside the main centres in the southeast and south becoming increasingly represented in the overall publication outcomes.

Brazilian scientific production has also switched in the past decade to focus more on original articles in leishmaniasis and CD. The number of reviews produced per year has steadily decreased which shows the maturity of Brazilian research in the area. Numbers of articles with international collaborators have also increased, following a global trend. This increase has been associated with the ease of data sharing, article access and communication between groups with the use of the internet that is now a central requirement for research groups throughout the world. International collaborations in Brazil are still focused on the Southeast region (Fig. 3) but this presented a noticeable shifting trend in the past three years, with a significant rise in international collaborations, particularly in the Northeast, which has demonstrated consistent growth in this area.

Efforts directed toward the internationalisation of research, such as those promoted by the National Graduate Plan (PNPG) and specific programmes like Science Without Borders (CsF) and the Institutional Internationalisation Programme (PrInt), may have had a significant impact on the expansion of scientific collaborations. These programmes facilitated academic mobility and international cooperation, leading to a substantial increase in the number of joint publications.[9,20] The establishment of the CAPES International Relations Directorate and the PrInt Programme, along with the increase in resources and personnel dedicated to internationalisation, created a more conducive environment for external collaborations and international publications. Although these programmes had a substantial impact on the national level, they also highlighted regional inequalities, as most of the approved projects were concentrated in the South and Southeast regions. The concentration of granted institutions in the South and Southeast emphasise the need for policies to reduce regional disparities.[9]

The publication impact of trypanosome research in Brazil has also increased, with most articles belonging to the Q1 quartile. The Southeast region is also the region in Brazil with the largest number of articles classified as Q1 by Qualis (Fig. 4), particularly in the field of leishmaniasis, which has a higher proportion of international publications compared to *T. cruzi*. Consequently, the citation impact of papers with international co-authors is generally higher than that of works published solely by Brazilian authors.[18] Additionally, research incentive programme like the PNPG and internationalisation projects, while not entirely successful in mitigating regional disparities, have contributed to the increasing number of Q1 publications in the northeast for *Leishmania* and in the north for both *Leishmania* and *T. cruzi*.

Brazil's contributions to tropical disease research, particularly in NTDs like CD and leishmaniasis, are significant and globally recognised.[21] These achievements are the result of historical milestones and the sustained efforts in research, public policy, and investment. The decentralisation of scientific production in Brazil, driven by national policies like the PNPG and the REUNI programme, has facilitated increased research output in historically underrepresented regions like the North and Northeast, that are the most affected by the NTDs.[22,23] Despite challenges such as fluctuating federal funding, the steady growth in publications, particularly original research articles, underscores the resilience of the Brazilian scientific community in the area. The success of these efforts is evident in Brazil's growing global influence, as seen in its leadership in trypanosome research and its increasing participation in international collaborations. Ensuring the continuity of these investments and initiatives is crucial for maintaining and expanding Brazil's role in tropical disease research, with long-term benefits for both national and global public health.

In addition to academic output, scientific research on trypanosomatids has a direct impact on public health. The Cuida Chagas Project, coordinated by the National Institute of Infectious Diseases (INI) of the Oswaldo Cruz Foundation (Fiocruz) in partnership with Bolivia, Brazil, Colombia, and Paraguay, and their respective Ministries of Health, combines implementation research and innovation. It aims to eliminate congenital transmission of CD by expanding access to diagnosis, treatment, and care, as well as validating diagnostic algorithms and therapeutic options under accessible conditions.[24] Another example is the standardisation of a molecular test for cutaneous leishmaniasis by Fiocruz in partnership with the Pan American Health Organisation (PAHO). This test was validated in multiple centres across Latin America and incorporated as a protocol into public health systems.[25] These initiatives demonstrate that scientific production has a direct impact on health policy formulation and the incorporation of health technologies, contributing to the fight against NTDs.

## ACKNOWLEDGEMENTS

To the GLORIOSOS clusters, at Programa Interunidades de Pós-Graduação em Bioinformática - Universidade Federal de Minas Gerais, that is managed by Izabela Mamede, Lúcio Queiroz and Herón Hilário.

## AUTHORS' CONTRIBUTION

BM - data collection, analysis, writing, and editing of the final version; IM - data analysis and final version editing; JRM, AR and AGBM - data collection and data analysis; ASMJ, ARN, ACSN, BCR, DOS, DO, FGCO, JD, LDCS, WRRF and DDLS - data collection; AMM - data analysis and financial support; GRF and CRM - data analysis, editing, and financial support. Authors declare no conflict of interest.

## DATA AVAILABILITY

The contents underlying the research text are included in the manuscript.

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

# OPEN PEER REVIEW

Memórias do IOC thanks the anonymous reviewers for their contribution to the peer review of this work.

## FIRST REVIEW ROUND

### REVIEWERS' COMMENTS

### REVIEWER #1

This is an interesting analyzes of Brazilian publication in the field of trypanosoma in the period of 2010-2021. Authors show a spreading of the scientific production through the country and an increment of international collaborations and explaining these facts with support politics. This is an important analysis that stimulates important discussions about scientific stimulus.

There are, however, some points to be considered:

Pg 6/35: Lines 20-22: It would be important to include some analysis showing the direct impact of the scientific production onto public health politics.

Pg 10/35 Line 52: It seems (analysing Figure 1A) that production in T. cruzi field is lower than 50%.

Pg 11/35 Lines 50-52: authors infer that the spread of publications is due the spread of scientific groups. However, some data should be presented to support this hypothesis.

Pg 12/35 Lines 24-26: Xs need to be replaced by data.

### AUTHORS' RESPONSE TO THE REVIEWERS

We appreciate the reviewer's careful reading and the valuable suggestions provided. The following changes have been made:

Pg 6/35, Lines 20–22: "It would be important to include some analysis showing the direct impact of the scientific production onto public health politics."

We have added data relating the impact of scientific production on public health policies at the end of the Discussion section.

Pg 10/35, Line 52: "It seems (analysing Figure 1A) that production in T. cruzi field is lower than 50%."

This has been corrected in the manuscript.

Pg 11/35, Lines 50–52: "Authors infer that the spread of publications is due to the spread of scientific groups. However, some data should be presented to support this hypothesis."

A new supplementary figure (Figure 2S) has been included to support this hypothesis.

Pg 12/35, Lines 24–26: "Xs need to be replaced by data."

The correction has been made.

## SECOND REVIEW ROUND

### REVIEWERS' COMMENTS

### REVIEWER #1

No comments.

