## [Reviewer Report · FIRST REVIEW ROUND - REVIEWERS COMMENTS]

## REVIEWER #1

This is an interesting analyzes of Brazilian publication in the field of trypanosoma in the period of 2010-2021. Authors show a spreading of the scientific production through the country and an increment of international collaborations and explaining these facts with support politics. This is an important analysis that stimulates important discussions about scientific stimulus.

There are, however, some points to be considered:

Pg 6/35: Lines 20-22: It would be important to include some analysis showing the direct impact of the scientific production onto public health politics.

Pg 10/35 Line 52: It seems (analysing Figure 1A) that production in T. cruzi field is lower than 50%.

Pg 11/35 Lines 50-52: authors infer that the spread of publications is due the spread of scientific groups. However, some data should be presented to support this hypothesis.

Pg 12/35 Lines 24-26: Xs need to be replaced by data.

---

## [Author Response · AUTHORS RESPONSE TO REVIEWERS]

## AUTHORS’ RESPONSE TO THE REVIEWERS

We appreciate the reviewer’s careful reading and the valuable suggestions provided. The following changes have been made:

Pg 6/35, Lines 20–22: “It would be important to include some analysis showing the direct impact of the scientific production onto public health politics.”

We have added data relating the impact of scientific production on public health policies at the end of the Discussion section.

Pg 10/35, Line 52: “It seems (analysing Figure 1A) that production in T. cruzi field is lower than 50%.”

This has been corrected in the manuscript.

Pg 11/35, Lines 50–52: “Authors infer that the spread of publications is due to the spread of scientific groups. However, some data should be presented to support this hypothesis.”

A new supplementary figure (Figure 2S) has been included to support this hypothesis.

Pg 12/35, Lines 24–26: “Xs need to be replaced by data.”

The correction has been made.